# Acculturation, Adaptation, and Health among Croatian Migrants in Austria and Ireland: A Cross-Sectional Study

**DOI:** 10.3390/ijerph192416960

**Published:** 2022-12-16

**Authors:** Izolda Pristojkovic Suko, Magdalena Holter, Erwin Stolz, Elfriede Renate Greimel, Wolfgang Freidl

**Affiliations:** 1Institute of Social Medicine and Epidemiology, Medical University of Graz, 8010 Graz, Austria; 2Institute for Medical Informatics, Statistics and Documentation, Medical University of Graz, 8036 Graz, Austria; 3Department of Obstetrics and Gynaecology, Medical University of Graz, 8036 Graz, Austria

**Keywords:** acculturation, adaptation, perceived health, quality of life, Croatian migrants

## Abstract

Since Croatia joined the European Union, majority of the studies on Croatian emigrants have predominantly addressed the reasons for migration and their future predictions. The primary purpose of this study was to investigate the relationship between the sense of coherence, health behavior, acculturation, adaptation, perceived health, and quality of life (QoL) in first-generation Croatian migrants living in Austria and Ireland. Our study is the first study that addresses the perceived health and QoL of Croatian migrants since the last emigration wave in 2013. An online survey was conducted in Austria (n = 112) and Ireland (n = 116) using standardized questionnaires. Multiple linear regression analyses were conducted for emigrated Croats to identify the predictors of perceived health and QoL. The analyses revealed that the sense of coherence and psychological adaptation were the strongest predictors of perceived health and QoL in Austria and Ireland. Furthermore, in the environmental domain of QoL, a higher education, higher net income, life in Austria rather than Ireland, better health behavior, higher sense of coherence, and better psychological and sociocultural adaptation explained 55.9% of the variance. Health policies and programs should use the salutogenic model to improve the health-related quality of life and psychological adaptation of Croatian migrants.

## 1. Introduction

A trend of increased emigration was observed since the Republic of Croatia became a European Union member state in 2013. Aside from Germany, Austria and Ireland have been the main destination countries for Croatian emigrants. In 2014, 6036 people emigrated from Croatia to Austria. Since 2015, these numbers have been slightly declining but still stood above 5000 per year. Compared with Austria, the number of Croatian emigrants in Ireland has been growing since 2013, peaking in 2016 at 5312 emigrated Croats. Since then, the emigration trend began to slightly decline.

Migration to a foreign country is a stressful life event. During migration, a person leaves their existing social position and adapts to a different lifestyle. Therefore, this event can create various mental pressures that might affect migrants’ physical and mental health [1]. In addition, the risk behavior of migrants can negatively influence their health. A recent systematic review showed that differences in the immigrant population composition and receiving country’s contexts predict the level and direction of changes in patterns of health risk behaviors. Moreover, the authors also pointed out that health risk behaviors among the migrant population are associated with acculturation [2].

Acculturation is defined as the changes in values, identity, beliefs, or changes in behaviors such as customs, diet, language, or social relationships [3]. These changes occur when individuals come in contact with a different culture [4]. During the process of acculturation, migrants develop a relationship with the new culture and maintain their original culture [5]. Thus, the two cultures come in contact, and both cultures may experience some change; in reality, one cultural group often dominates the other [6].

Initially, the process of acculturation was conceptualized as unidimensional, in which it is expected that the migrants acquire the values, beliefs, and practices of their new home country and discard those from their cultural background [7]. Nowadays, acculturation is often conceptualized as multidimensional because it is influenced by several contextual factors and because both cultures go through changes due to the influence of the other [5,8]. Hence, measurements of acculturation can be conceptualized in three ways: namely unidimensional, bidimensional, and multidimensional [9].

Berry developed a categorical model of acculturation based on the extent to which the culture of origin is being maintained and the extent to which the new host culture is adopted [10]. His model measures acculturation orientation on two independent scales and allocates participants based on their scores to one of the four acculturation strategies by splitting the scores of orientation toward the home and host culture into four quadrants [11]. However, other authors [11,12,13] have critically analyzed the existing acculturation measures and recommended the bidimensional approach, which measures home and host country orientation as independent and continuous variables. Therefore, this approach was used for this study.

Acculturation orientations facilitate cultural adjustment, i.e., adaptation [14]. Ward and Kennedy classified adaptation outcomes into psychological and sociocultural adaptation [15]. Psychological adaptation involves subjective psychological and physical well-being as well as life satisfaction, whereas sociocultural adaptation comprises the acquired social skills necessary to “fit in” with a new cultural context and an individual’s competence to organize daily life [16].

The Multidimensional Individual Difference Acculturation (MIDA) model developed by Safdar et al. [17] suggested that there are three predictor variables of acculturation attitudes and adaptation outcomes: psychosocial resources, co-national connectedness, and hassles. In this model, it is important to distinguish between in-group and out-group social support when predicting the migrants’ psychological well-being [17]. In addition, the MIDA model presents a framework that analyzes the acculturation process of individuals, thereby focusing on the core factors that can influence the adaptation process of immigrants in a new society [18].

Acculturation is also considered as an explanatory factor for health inequalities [19]. Furthermore, previous studies have also considered acculturation as a factor influencing migrants’ health in general [9]. However, the results from previous studies have shown inconsistencies considering the magnitude and direction of the effects; thus, general statements about this connection cannot be made [20]. Further research is necessary to examine the relevant migration-specific aspects that could help to explain health inequalities [9].

Another approach to explain the health status of migrants is Antonovsky’s salutogenic model. Its core concept, sense of coherence (SOC), focuses on the ability of individuals to cope with stressors in life while maintaining their health status [21]. There is empirical evidence that SOC determines a variety of health dimensions (e.g., physical health, mental health, well-being, life satisfaction, etc.) [22,23], shapes migrants’ acculturation process [24,25], and predicts health-related quality of life (QoL) [26,27]. The WHO defines QoL as a subjective evaluation of an individual’s position in life and thus includes physical health, psychological state, social relationships, level of independence, personal beliefs, and their relationship to features of the environment [28].

Previous studies on the immigrant population showed that socio-demographic variables, i.e., sex, age, marital status, education, and income, are associated with QoL [29,30]. In addition, studies have also shown that these variables affect acculturation and are positively associated with psychological and sociocultural adjustment [31,32,33]. However, previous research has shown conflicting results considering the effect of socio-demographic variables on SOC [34,35,36]. Nevertheless, Sardu et al. [37] found that these variables influence SOC and should be taken as confounders when comparing SOC values among different populations.

Riedel et al. combined Berry’s and Antonovsky’s models to provide different perspectives on both the formation and effects of acculturation strategies that trigger psychological acculturation processes and to explain the relationship between migration and mental health [38]. The authors highlighted the lack of evidence regarding the implications of the SOC on acculturation processes and psychological adaptation, which holds important practical suggestions for primary health care [38].

The differences between the quality and costs of health care in the countries of origin and destination can also affect migrants’ health. Although there are differences between Austria, Ireland, and Croatia concerning the healthcare system, all three countries offer adequate healthcare. However, a migration background can make access to these healthcare systems difficult, and many migrants struggle to access healthcare due to several access barriers, which can harm their health.

In a recent systematic review, the health and access to healthcare of migrants and non-migrants were compared. The authors concluded that inequalities persist due to barriers such as language skills, level of health literacy, awareness of healthcare systems, and means to pay for health services [39]. Furthermore, there are differences in health policies across EU member states, and not all of them properly address the health of migrants [40]. To reduce obstacles in healthcare access, research that highlights the views of the migrants on their own health and migrant patient experience is important [39,41].

The results of previous research give an overview of how many factors influence migrants’ health. To date, the majority of studies on Croats only shed light on the motives for emigration [42,43], and very little is known about their adaptation to the new environment as well as their perceived health and QoL in the host country. Therefore, the present study aimed to determine the associations of acculturation, adaptation, SOC, and health behavior with the perceived health and QoL of first-generation Croatian migrants. Furthermore, we investigated the extent to which perceived health and the four domains of QoL were predicted by sociodemographic variables, health behavior, SOC, acculturation, and psychological and sociocultural adaptation.

Based on the theoretical background, the following hypotheses were formulated:

**Hypothesis** **1.***Sociodemographic variables, health behavior, SOC, acculturation, and psychological and sociocultural adaptation are significant predictors of perceived health and QoL of first-generation Croatian migrants*.

**Hypothesis** **2.***Acculturation outcomes (i.e., psychological and sociocultural adaptation), SOC, and sociodemographic characteristics are positively associated with the perceived health and QoL of first-generation Croatian migrants*.

**Hypothesis** **3.***There are significant differences between Croats in Austria and Ireland concerning acculturation, adaptation, perceived health, and QoL*.

To the best of our knowledge, this is the first study evaluating the above-mentioned parameters among Croatian migrants living in Austria and Ireland.

## 2. Materials and Methods

### 2.1. Study Design

This was a cross-sectional, questionnaire-based cross-cultural study. Using the LimeSurvey tool, an online survey was conducted with comparison groups in Austria and Ireland. The survey was conducted in two cities (Graz and Dublin) between October 2019 and April 2020 (n = 228). The response data have been stored in a separate database with a username/password for the LimeSurvey Cloud instance. Only the authors have access to these data. All measures were carried out following the Good Scientific Practice guidelines.

Furthermore, in Graz, Austria, a small number of participants were personally recruited throughout the Croatian community. These participants either obtained the link for the online questionnaire or filled out the questionnaire in printed form.

### 2.2. Study Population

Study participants in Graz and Dublin were first-generation immigrants between 20 and 55 years old who have been living in the immigration country for a period between ten months and five years. The criterion of ten months was selected as a previous study on Croatian immigrants showed that in this length of time, immigrants acquire sufficient linguistic competence in the host country [44].

The study participants were divided into three age groups, with the first group being 20–31 years old (40%), the second being 32–43 years old (40%), and the third being 44–55 years old (20%); this was chosen based on the previous research, where the majority of migrated Croats were between 25 and 40 years of age [43]. In addition, the fulfilment of quotas took place according to age and sex (quota of 50% each). The fulfilment of the quotas is presented in Table 1. Participation was voluntary and anonymous. By filling in the survey, participants gave their full consent.

In both countries, the data were simultaneously collected. The recruitment process occurred online, mainly through social networks (Facebook, Instagram, and LinkedIn) as well as via e-mail.

### 2.3. Measurements

The questionnaire covered the following topics: sociodemographic characteristics, health behavior, somatic symptoms, SOC, QoL, and scales for acculturation orientations and adaptations. Standardized questionnaires were used. Five scales (Simple Lifestyle Indicator Questionnaire—SLIQ, Somatic Symptom Scale-8—SSS-8, Brief Acculturation Orientation Scale—BAOS, Brief Psychological Adaptation Scale—BPAS, and Brief Sociocultural Adaptation Scale—BSAS) were translated into Croatian by bilingual speakers and back-translated. The translations were compared with the original version and discussed to avoid translation errors.

To verify the translation of the questionnaires and to determine whether the factor structure from the literature can be found in the translations, a confirmatory factor analysis was conducted. The number of factors in each model depended on the questionnaires’ original structure (5 factors for SLIQ, 4 factors for SSS, 2 factors for BAOS, 1 factor for BPAS, and 1 factor for BSAS 1). More information on the structure of each questionnaire can be obtained from the questionnaires’ publications.

In order to investigate the models’ fit, the root mean square error of approximation (RMSEA), Tucker–Lewis index (TLI), and comparative fit index (CFI) were calculated. RMSEA < 0.05 was considered a good fit, 0.05 < RMSEA < 0.08 was an adequate fit, and RMSEA > 0.08 was a bad fit [45]. In addition, the CFI and TLI values also ranged from 0 to 1, with larger values indicating a better fit [45]. After estimating the models, the fit statistics were obtained. As can be seen in Table 2, all of the models showed an adequate fit, except for BPAS.

For BPAS, as in the literature, one factor was assumed, on which the eight items were loaded. The data fit the latent factor. However, the items did not seem to load on a factor as intended in the literature, and a comparison of the factor structure in other studies could not be found. The poor model fit can be explained by the fact that some response categories were only selected by very few people. Furthermore, the low n = 228 (Austria and Ireland) is a further explanation.

To assess the reliability of the questionnaires, the Cronbach alpha was calculated. Values of 0.70 or higher indicated an acceptable internal consistency [46]. The scales showed acceptable reliability, whereby three scales (BAOS-home, BPAS, and BSAS) showed good reliability (Table 3).

Sociodemographic characteristics included sex, age, length of stay (in months), level of education (secondary or tertiary), living situation (living alone or with a partner), occupation status, and net household income.

The brief version of the World Health Organization Quality of Life (WHOQOL-BREF) assessment instrument was used to measure the quality of life. It is a valid, cross-culturally applicable questionnaire comprising 24 questions within four domains: physical health, psychological health, social relationships, and environmental health [47]. In addition, it consists of two separate questions about the individuals’ overall perception of quality of life and general health. Participants rated their agreement with each item on a 5-point Likert-type scale, with higher scores indicating better quality of life. The total score ranges from 0 to 100. High internal consistency and test–retest stability were established for the results in the Croatian version of the WHOQOL-BREF [48].

Health behavior was assessed using the Simple Lifestyle Indicator Questionnaire (SLIQ). The questionnaire contains 5 components: diet (3 questions), activity (3 questions), alcohol consumption in an average week (3 questions), smoking habits (2 questions), and stress in everyday life (1 question). The raw and categorical scores were calculated for each component. Overall, the SLIQ score can range from 0 to 10 because each component has a category score of 0, 1, or 2 [49]. The higher the score, the more healthy the lifestyle. One question related to smoking was added to assess the average number of cigarettes, cigars, pipes, or other tobacco products smoked per day among smokers.

The Somatic Symptom Scale-8 (SSS-8) is a brief self-reported measure used to assess the somatic symptom burden. Respondents rated their somatic symptoms, i.e., gastrointestinal, pain, fatigue, and cardiopulmonary, within the last seven days on a 5-point response scale ranging from not at all (0) to very much (4) [50]. The simple sum score can range from 0 to 32. Higher scores indicate a higher somatic symptom burden. The internal consistency was between 0.77 and 0.93 for the different language versions [51].

The 13-item Sense of Coherence Scale (SOC-13) was used to measure the sense of coherence. The SOC-13 scale is a valid and reliable instrument [52]. The scale has been validated in the Croatian population [53]. It consists of three subscales: comprehensibility (5 items), manageability (4 items), and meaningfulness (4 items). Participants rated their agreement with each item on a 7-point Likert scale. The total score ranges from 13 to 91, whereby higher scores indicate a stronger SOC.

Questionnaires developed by Demes and Geeraert [12] were used to measure acculturation and psychological and sociocultural adaptation. The Brief Acculturation Orientation Scale (BAOS) was used to independently measure the acculturation orientation to the home country (BAOS-Home) and host country (BAOS-Host). Participants were asked to rate their agreement about the four central indicators of acculturation orientation (traditions, friendship, characteristics, and actions) on a 7-point Likert-type scale ranging from 1 “strongly disagree” to 7 “strongly agree”.

The Brief Psychological Adaptation Scale (BPAS) was used to measure psychological adaptation. The scale contains a list of 8 items about positive and negative feelings toward the home and host countries. Participants were asked to “Think about living in [host country]. In the last two weeks, how often have you felt…” and respond on a 7-point Likert-type scale (1 “never” and 7 “always”), to items such as “Frustrated by difficulties adapting to [host country]” [12].

Sociocultural adaptation was assessed using the Brief Sociocultural Adaptation Scale (BSAS). It consists of 12 items. Participants were asked to “Think about living in [host country]. How easy or difficult is it for you to adapt to…” and then rate the items on a 7-point Likert-type scale from 1 “very difficult” to 7 “very easy” [12].

### 2.4. Statistical Analysis

Frequencies were run on the data to discover any missing values or outliers. In addition, the data were checked regarding the normal distribution. Data analysis included descriptive statistics such as mean and standard deviation, which were used to analyze the variables by country. An independent group *t*-test was conducted to compare the means between Austria and Ireland. Correlations between BAOS-Home, BAOS-Host, BPAS, and BSAS were calculated, and a regression approach was used to assess the relationship between acculturation and adaptation in first-generation Croatian migrants. Furthermore, a multivariable linear regression analysis was used to assess the association between variables and determine the predictors of perceived health and quality of life in Austria and Ireland. Different domains of QoL (physical health, psychological health, social relationships, and environment) and perceived health (somatic symptoms and perception of overall general health) were separately included as dependent variables. Sex, age, education, living situation, net household income, health behavior, country, length of stay, SOC, BAOS-Home, BAOS-Host, BPAS, and BSAS were included as predictor variables. All nominal and ordinal variables, i.e., sex, education, living situation, and country, were dummy-coded. A series of tests, e.g., linear relationship, multivariate normality, multicollinearity, and homoscedasticity, were conducted to check for assumptions for linear regression. A *p*-value < 0.05 was considered statistically significant.

Furthermore, the dependent variable “Perception of overall general health” was dichotomized (0—not satisfied, 1—satisfied), and a binary logistic regression was calculated. Therefore, for this dependent variable, instead of the adjusted R^2^, the Nagelkerke R^2^ value was reported. Moreover, the odds ratios (OR) with 95% confidence intervals (CI) were calculated.

Sociodemographic variables and health behavior were considered possible confounders. First, an unadjusted regression model was calculated only using the central predictors, i.e., SOC, BAOS-Home, BAOS-Host, BPAS, and BSAS. Then, an adjusted regression model with the confounders and central predictors was calculated. Finally, the delta of the R^2^ value between the unadjusted and adjusted models was calculated to determine what contribution these predictors made to our assumptions.

Descriptive statistics and multiple linear regression models were calculated using SPSS statistical software (version 25.0; IBM Corp, Armonk, NY, USA). The analyses of the factorial structure of the questionnaires were performed with R software (version 4.0.3) using the packages “Multidimensional Item Response Theory “mirt” [54] and “lavaan” [55].

## 3. Results

### 3.1. Sociodemographic Data

The sample comprised 228 Croats living in Austria (112) and Ireland (116). It consisted of 141 females (61.8%) with an average age of 33.89 years and 87 males (38.2%) with an average age of 33.77 years. The average time spent abroad for Croats in Austria was 39.05 months (SD = 17.18); for Croats in Ireland, it was 35.19 months (SD = 15.58). The majority of the participants in both countries lived with their partners (58.9% Croats in Austria vs. 67.2% Croats in Ireland). The socioeconomics data are presented in Table 4.

### 3.2. Acculturation and Adaptation

Regarding acculturation orientation, the correlation between BAOS-Home and BAOS-Host scores was low but still statistically significant (r = 0.24, *p* < 0.001). The results suggested that participants weakly tended to orient themselves towards one culture if they were more oriented to the other culture. The correlation between BPAS and BSAS scores showed a strong and statistically significant correlation, with r = 0.62, *p* < 0.001.

Furthermore, psychological and sociocultural adaptation were separately regressed on the BAOS subscale. Home and host orientation scores accounted for 13.6% of the variance in the BPAS, with home orientation being negatively related to BPAS (β = −0.34, *p* < 0.001); meanwhile, the host orientation was positively related to BPAS, with β = 0.27, *p* < 0.001.

Similar results were observed in case of the BSAS. Home and host orientation accounted for 13.3% of the variance in the BSAS. Again, home orientation was negatively related, with values of β = −0.13 and *p* = 0.044, and the host orientation was positively related to BSAS (β = 0.38, *p* < 0.001). This showed a negative relationship between home orientation and adaptation and a positive relationship between host orientation and adaptation.

### 3.3. Perceived Health and Quality of Life

The comparison of health behavior, SOC, acculturation, adaptation, perceived health, and QoL between Croats living in Austria and Ireland is shown in Table 5. Croats living in Ireland were more psychologically adapted compared with those in Austria. These showed significantly higher mean scores on the BAOS-Home subscale as well as the environmental health domain of QoL.

The results of the linear regression analysis showed a statistically significant association between the predictor variables with the perceived health and all four domains of QoL. In this study, two linear regression models were calculated, whereby the second model was adjusted for the possible confounders, i.e., sociodemographic variables and health behavior (Table 6 and Table 7). In summary, the defined central predictors were the same in the unadjusted and adjusted linear regression models.

The SOC and psychological adaptation were the strongest predictors of the perceived health and the four domains of QoL in Austria and Ireland. In addition, sociodemographic variables and health behavior contributed the most to the explanation of the environmental health domain (delta R^2^ = 11.9%).

In both regression models, significant predictors of somatic symptoms were a lower SOC, higher home orientation, and lower psychological adaptation. The SOC was a significant predictor of the perception of overall general health in both regression models. In addition, net household income was positively associated whereas the length of stay was negatively associated with the perception of the overall health in the adjusted model.

For the physical health domain of QoL, significant predictors were a higher SOC and higher psychological adaptation. Furthermore, age was negatively associated with this domain of QoL in the adjusted model. Both unadjusted and adjusted models showed that a higher SOC, lower home orientation, and better psychological adaptation were associated with the psychological health domain of QoL. The SOC, home orientation, and psychological adaptation were positively associated with the social relationship domains of QoL.

In the environmental health domain of QoL, a higher SOC and better psychological and sociocultural adaptation were identified as significant predictors explaining for 44% of the variance in the unadjusted regression model. In the adjusted regression model, the significant predictors were a higher education, higher net income, life in Austria, better health behavior, higher SOC, and better psychological and sociocultural adaptation, explaining for 55.9% of the variance.

## 4. Discussion

In our study, Croatian migrants in both countries were comparable regarding their sociodemographic status. However, among Croats living in Ireland, women showed more interest in participating in the study, and almost two-thirds of the participants were female. Moreover, Croatian migrants in Ireland had a higher net household income.

Home and host country orientation were measured on continuous scales and examined using a regression approach. Consistent with prior research [12], the BPAS and BSAS scores were positively correlated. In line with expectations, our results indicated that migrants who are more oriented toward the host culture adapt better than those who are more oriented toward the home culture. Kosic [44] also concluded that Croatian and Polish immigrants living in Italy who feel accepted and have positive attitudes toward the host country are psychologically and socioculturally well-adapted. Willingness to learn about the new culture is one of the aspects of adaptation. If the migrants surround themselves with people from their host country, they can learn more about the host culture and learn the new language quicker than those migrants who only socialize with people from their home country. Furthermore, the psychological and sociocultural adaptation of migrants are influenced by other external or host-country factors, such as job opportunities or migrant-friendly healthcare environments.

Our results showed differences between Croatian migrants living in Austria and Ireland concerning acculturation, adaptation, and QoL. Croatian migrants living in Ireland, compared with those living in Austria, had higher mean scores for psychological adaptation. Psychological adaptation is an element of general satisfaction with life in the host environment [56]. Based on this outcome, Croats in Ireland are better psychologically adapted and more satisfied with life in the host country, which indicates that they tend to orient themselves toward the host culture. In line with our study, a study among Croatian migrants in Dublin [57] indicated that Croatian migrants are open to accepting and learning about Irish culture, which is a good foundation for successfully adapting to the host country.

On the other hand, our results showed that Croats living in Austria are more oriented toward their home country. This may be explained by the fact that Austria, especially Graz where a lot of participants live, is geographically closer to Croatia, and these migrants have the opportunity to drive home more often. Language skills can also be one of the reasons. Croats learn English from an early age because, for example, in Croatia, English movies are not synchronized. Compared to English, German is taught in school and is not often used in daily life. Therefore, Croats in Austria face more challenges while learning German, which could suggest that they socialize more with other Croats.

Another difference between Croatian migrants in Austria and Ireland is that those living in Austria showed higher mean scores on the environmental health domain of the QoL. The environmental health domain of the QoL contains facets on financial resources, safety, and security, accessibility of healthcare services, and physical environment [58]. This outcome can be explained by the existing differences between Austria and Ireland, e.g., accessibility and quality of health and social care, housing, transport, etc. For instance, in Austria, 99.9% of the population is insured based on the compulsory state-funded healthcare [59], whereas in Ireland, only about 32% of the population has a medical card that entitles the holders to free hospital care, general practitioner (GP), dental services, prescription drugs, etc. [60]. The rest of the population is not entitled to a medical card and has to pay for a number of health care services. In addition, Austrian cities, compared to Irish ones, have a less commercialized housing market and a more active social housing policy [61]. Although Croatian migrants in Ireland have higher monthly net household income compared with those in Austria, living costs in Ireland are also higher, which can directly affect the outcomes of this domain of QoL. The environmental health domain of the QoL plays a crucial role for migrants because, especially in the beginning, they are often affected by social inequalities.

In this study, the common predictors of perceived health and QoL in the linear regression model for Austria and Ireland were identified. The SOC is the strongest predictor of perceived health and all QoL domains. This is consistent with a previous study among immigrants [62], which indicated that SOC is positively associated with QoL. Another significant predictor of somatic symptoms and all QoL domains was the psychological adaptation, which was positively associated with the domains of QoL and negatively associated with the somatic symptoms. Except for the SOC and psychological adaptation, the home orientation was another factor that determined somatic symptoms. The data indicated that participants who were more oriented toward home culture exhibited greater somatic symptoms.

Among the central predictors, only SOC and psychological adaptation predicted all QoL domains, whereas other variables predicted some of the domains. Contrary to our expectations, host orientation was not identified as a predictor of perceived health and QoL. However, in the environmental domain of QoL, psychological as well as sociocultural adaptation were significant predictors, which can be explained by the fact that both adaptations are linked to the host society.

In the adjusted linear regression model, several sociodemographic variables were predictors of the QoL. Length of stay was negatively associated with the perception of overall general health, which is consistent with the previous study on the migrant population [63]. Regarding the physical health domain of QoL, higher age was a significant predictor of lower physical health, which is in line with the previous research [64]. In the social relationships domain of QoL, the living situation, more precisely living with the partner, was a significant predictor. This outcome is in line with a study, which showed that living together with the partner and thus being married is a predictor of a higher QoL [65]. In the environmental domain of QoL, sociodemographic variables and health behavior contributed the most to the explanation of the model compared to other domains of QoL. Furthermore, higher education and higher net income were predictors of this domain of QoL. This is consistent with a previous study on the migrant population, which showed that higher education and income contribute to an improved QoL [66].

Established scales that had good internal consistency and acceptable Cronbach alpha values were used in this study. In addition, the predictors of perceived health and QoL among Croats living in two different countries were defined. The SOC is the strongest predictor of perceived health and QoL, and it can therefore function as an indicator of the health status of first-generation Croatian migrants. As shown in the literature, the SOC not only predicts the perceived health and QoL but also allows the migrants with a strong SOC to cope with stressors during the migration and adapt faster to the new environment. Similarly, our results also showed that psychological adaptation is another important predictor of perceived health and QoL. Instead of solely focusing on health behavior change, as has been performed by many conventional programs, salutogenic components could be included in health promotion programs targeting migration populations.

### Limitations

Some limitations of this study should be mentioned. Although attempts have been made to recruit participants in all age groups and fulfill the quotas accordingly, not all quotas were fulfilled as planned. In the last age group (44–55 years), 81.8% of the participants were females. First, one reason for this might be that women were more active than men in the Facebook groups where the link was posted. Secondly, many questionnaires filled by men have been excluded because they were not appropriately filled out. Some answers were illogical, particularly in the question relating to the number of drinks consumed in the average week. Finally, access to the target group was mainly possible through social networks, which might have led to selection bias.

Because the survey was conducted online, another limitation is the possibility that Croatian migrants not living in Graz or Dublin might have taken part in the survey.

The last limitation is the cross-sectional design of this study. Although the findings of this study contributed to the understanding of perceived health and QoL among Croatian migrants, a longitudinal design is highly desirable in order to measure how perceived health and QoL change over time.

## 5. Conclusions

In this study, it can be stated that a higher SOC and higher psychological adaptation are significantly associated with higher perceived health and QoL in Austria and Ireland. To the best of our knowledge, our cross-sectional study is the first study that has had several hundred participants and sheds light on the perceived health and quality of life, which is important for the health monitoring and health promotion of Croatian migrants in Austria and Ireland.

Overall, knowledge about the predictors of perceived health and QoL can minimize the barriers to accessing health services, improve the delivery of preventive care, and provide a basis for health policies to guarantee the appropriate healthcare for a growing number of Croatian immigrants. Furthermore, it can be used to promote health-related quality of life and psychological adaptation for Croats living abroad.

## Figures and Tables

**Table 1 ijerph-19-16960-t001:** Fulfilment of the quotas according to age and sex.

		Austria	Ireland
20–31 years	Female	33 (64.7%)	28 (57.1%)
	Male	18 (35.3%)	21 (42.9%)
	Total	51	49
32–44 years	Female	22 (52.4%)	30 (66.7%)
	Male	20 (47.6%)	15 (33.3%)
	Total	42	45
45–55 years	Female	10 (52.6%)	18 (81.8%)
	Male	9 (47.4%)	4 (18.2%)
	Total	19	22

**Table 2 ijerph-19-16960-t002:** Confirmatory factor analysis.

	RMSEA (95% CI)	TLI	CFI
SLIQ	0.059 (0.046–0.072)	0.875	0.896
SSS-8	0.093 (0.072–0.115)	0.862	0.901
BAOS	0.095 (0.068–0.123)	0.936	0.954
BPAS	0.266 (0.241–0.29)	0.639	0.742
BSAS	0.144 (0.129–0.16)	0.894	0.913

**Table 3 ijerph-19-16960-t003:** Reliability coefficients of the instruments.

Abbreviations	Scale	No. of Items	Range	M	SD	Cronbach α
WHOQOL-BREF	World Health Organization Quality of Life Scale	26				0.908
	Physical Health	7	6–20	16.01	2.31	0.783
	Psychological Health	6	7–20	15.05	2.48	0.786
	Social Relationships	3	7–20	15.35	3.03	0.669
	Environment	8	7–20	14.82	2.48	0.784
SLIQ	Simple Lifestyle Indicator Questionnaire	12	0–10	5.88	1.71	
SSS-8	Somatic Symptom Scale	8	0–23	6.14	4.75	0.796
SOC	Sense of Coherence	13	35–71	55.76	6.07	0.799
BAOS	Brief Acculturation Orientation Scale					
BAOS-Home	Home Orientation	4	4–28	15.53	6.52	0.852
BAOS-Host	Host Orientation	4	4–28	17.30	5.04	0.788
BPAS	Brief Psychological Adaptation Scale	8	18–53	40.13	8.02	0.851
BSAS	Brief Sociocultural Adaptation Scale	12	33–84	65.05	12.63	0.894

**Table 4 ijerph-19-16960-t004:** Sociodemographic characteristics of the participants.

	Austria n = 112 n (%)	Ireland n = 116 n (%)
Sex		
Female	65 (58%)	76 (65.5%)
Male	47 (42%)	48 (34.5%)
Age		
20–31	51 (45.5%)	49 (42.2%)
32–44	42 (37.5%)	45 (38.8%)
45–55	19 (17%)	22 (19%)
Living situation		
Living with partner	66 (58.9%)	78 (67.2%)
Living alone	46 (41.1%)	38 (32.8%)
Education		
Secondary	61 (54.5%)	59 (50.9%)
Tertiary	51 (45.5%)	57 (49.1%)
Net household income in EUR		
Up to 1.300	20 (17.9%)	4 (3.4%)
Up to 2.500	44 (39.3%)	30 (25.9%)
Up to 4.000	41 (36.5%)	42 (36.2%)
Above 4.500	7 (6.3%)	40 (34.5%)
Length of stay (in months)		
10–12	9 (8.0%)	5 (4.3%)
13–24	19 (17.0%)	32 (27.6%)
25–36	19 (17.0%)	23 (19.8%)
37–48	19 (17.0%)	22 (19.0%)
49–61	46 (41.0%)	34 (29.3%)

**Table 5 ijerph-19-16960-t005:** Difference between Croats living in Austria and Ireland.

	Austria Mean (SD)	Ireland Mean (SD)	*p*	Cohen’s d
SLIQ	5.93 (1.92)	5.66 (1.67)	0.267	0.15
SOC	62.17 (13.12)	62.55 (13.52)	0.829	−0.03
BAOS-Home	17.36 (6.42)	13.77 (6.14)	<0.001	0.57
BAOS-Host	17.93 (4.79)	16.69 (5.23)	0.064	0.25
BPAS	38.89 (8.09)	41.32 (7.80)	0.022	−0.31
BSAS	65.66 (11.85)	64.46 (13.36)	0.473	0.09
SSS-8	6.61 (4.48)	6.22 (5.16)	0.551	0.08
Perception of overall health	3.88 (0.86)	3.81 (0.85)	0.570	0.08
QoL Physical health	15.96 (2.18)	16.14 (2.65)	0.580	−0.07
QoL Psychological health	15.08 (2.62)	15.08 (2.49)	0.993	0.00
QoL Social relationships	15.65 (2.96)	15.09 (3.09)	0.162	0.18
QoL Environment	15.47 (2.27)	14.26 (2.80)	<0.001	0.47

**Table 6 ijerph-19-16960-t006:** Associations between perceived health, quality of life, and predictor variables—unadjusted linear regression model.

					Quality of Life							
	Somatic Symptoms		Perception of Overall General Health		Physical Health		Psychological Health		Social Relationships		Environment	
Variables	β	*p*	OR	*p*	β	*p*	β	*p*	β	*p*	β	*p*
SOC	−0.371	<0.001	1.07 (1.04–1.11)	<0.001	0.288	<0.001	0.453	<0.001	0.180	0.008	0.203	0.001
BAOS-Home	0.129	0.044	1.01 (0.95–1.07)	0.819	−0.018	0.777	0.175	0.001	0.245	<0.001	0.047	0.400
BAOS- Host	−0.025	0.685	1.02 (0.95–1.10)	0.550	0.019	0.757	−0.002	0.969	−0.015	0.814	0.015	0.783
BPAS	−0.219	0.008	1.02 (0.96–1.08)	0.588	0.274	0.001	0.290	<0.001	0.282	0.001	0.209	0.004
BSAS	0.010	0.895	1.03 (0.99–1.06)	0.158	0.113	0.137	0.120	0.060	0.142	0.072	0.393	<0.001
Adjusted R^2^	27.1%		28.0% *		30.5%		51.5%		25.4%		44.0%	

* Nagelkerke R^2^.

**Table 7 ijerph-19-16960-t007:** Associations between perceived health, quality of life, and predictor variables—adjusted linear regression model.

					Quality of Life							
	Somatic Symptoms		Perception of Overall General Health		Physical Health		Psychological Health		Social Relationships		Environment	
Variables	β	*p*	OR	*p*	β	*p*	β	*p*	β	*p*	β	*p*
Sex ^(a)^	0.095	0.119	0.95 (0.44–2.07)	0.898	−0.107	0.068	−0.080	0.106	−0.022	0.719	−0.092	0.054
Age	0.058	0.357	0.94 (0.92–1.01)	0.122	−0.120	0.046	−0.022	0.657	−0.045	0.476	−0.088	0.071
Education ^(b)^	−0.021	0.739	0.73 (0.34–1.56)	0.410	−0.054	0.368	−0.011	0.831	−0.037	0.580	0.102	0.037
Living Situation ^(c)^	0.021	0.752	0.50 (0.21–1.23)	0.132	−0.111	0.089	0.052	0.338	0.139	0.044	−0.102	0.053
Net household income	−0.064	0.391	1.88 (1.12–3.17)	0.017	0.085	0.234	0.070	0.244	0.000	0.995	0.155	0.008
Country ^(d)^	−0.030	0.660	1.59 (0.70–3.61)	0.264	−0.003	0.965	0.017	0.750	0.084	0.219	0.281	<0.001
Length of stay	−0.084	0.174	0.97 (0.95–1.00)	0.025	−0.051	0.384	−0.080	0.107	−0.049	0.432	0.009	0.856
SLIQ	−0.044	0.487	1.13 (0.91–1.40)	0.262	0.064	0.290	0.128	0.013	0.044	0.491	0.101	0.041
SOC	−0.356	<0.001	1.08 (1.04–1.12)	<0.001	0.314	<0.001	0.413	<0.001	0.169	0.022	0.176	0.002
BAOS-Home	0.141	0.036	1.01 (0.95–1.08)	0.754	−0.006	0.931	0.159	0.003	0.202	0.003	0.015	0.777
BAOS- Host	−0.041	0.525	1.04 (0.96–1.13)	0.309	0.056	0.364	0.004	0.938	−0.022	0.741	0.020	0.687
BPAS	−0.210	0.015	1.01 (0.95–1.08)	0.748	0.252	0.002	0.250	<0.001	0.272	0.002	0.255	<0.001
BSAS	0.027	0.730	1.02 (0.98–1.06)	0.291	0.107	0.155	0.110	0.083	0.129	0.108	0.322	<0.001
Adjusted R^2^	27.2%		36.3% *		33.4%		52.8%		25.1%		55.9%	
Delta R^2^	0.1%		8.3%		2.9%		1.3%		−0.3%		11.9%	

* Nagelkerke R^2^. ^(a)^ female is a reference value. ^(b)^ tertiary education is a reference value. ^(c)^ living with partner is a reference value. ^(d)^ Austria is a reference value.

## Data Availability

All data generated or analyzed during this study are included in this published article and its Appendix A.

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
