# Peer review of "Acculturation, Adaptation, and Health among Croatian Migrants in Austria and Ireland: A Cross-Sectional Study"

_ijerph, 2022, doi:10.3390/ijerph192416960_

Round 1
Reviewer 1 Report
Dear authors,
Thank you very much for your contribution. Your work is really important for the interdisciplinarity required for the studies on public health for migrants' health and organization of health and care systems in EU.
However, working on conceptualizing different approaches and theories from different disciplines also carries a few challenges. Please, consider my comments in order to improve the quality of your manuscript or at least the understanding of future readers:
Regarding the conceptualization of acculturation in the manuscript, you might find useful the following paragraph to theoretically explain the concept in your manuscript:
“Acculturation is the process by which migrants to a new culture develop relationships with the new culture and maintain their original culture (Berry & Sam, 1997). Acculturation has been classically defined as the changes that develop when groups of individuals come into contact with a different culture (Redfield, Linton, & Herskovits, 1936). This process was initially conceptualized as unidimensional, in which retention of the original culture and acquisition of the new host culture were cast at opposing ends of a single continuum (Schwartz, Unger, Zamboanga, & Szapocznik, 2010). According to this unidimensional model, migrants were expected to acquire the values, practices, and beliefs of their new homelands and discard those from their cultural heritage. Acculturation is now more often conceptualized as complex and multidimensional, meaning that both cultures change under the influence of each other and acculturation is influenced by a number of contextual factors (Berry & Sam, 1997; Sam & Berry, 2006)” extracted from Tanenbaum, M.L., Commissariat, P., Kupperman, E., Baek, R.N., Gonzalez, J.S. (2013). Acculturation. In: Gellman, M.D., Turner, J.R. (eds) Encyclopedia of Behavioral Medicine. Springer, New York, NY. https://doi.org/10.1007/978-1-4419-1005-9_147
Bilbliography:
Berry, J. W., & Sam, D. L. (1997). Acculturation and adaptation. In J. W. Berry, M. H. Segall, & C. Kagitcibasi (Eds.), Handbook of cross-cultural psychology (2nd ed., Vol. 3, pp. 291–326). Boston: Allyn and Bacon.
Redfield, R., Linton, R., & Herskovits, M. (1936). Memorandum on the study of acculturation. American Anthropologist, 38, 149–152.
Sam, D., & Berry, J. W. (2006). Acculturation: Conceptual background. In D. Sam & J. W. Berry (Eds.), The Cambridge handbook of acculturation psychology (pp. 17–18). Cambridge: Cambridge University Press.
Schwartz, S. J., Unger, J. B., Zamboanga, B. L., & Szapocznik, J. (2010). Rethinking the concept of acculturation: Implications for theory and research. American Psychologist, 65, 237–251.
When explaining acculturation orientations and their relationship with variables such as psychological and cultural adjustment, psychological and physical well-being, it makes necessary to mention the Multidimensional Individual Model of Acculturation by Professor Saba Safdar.
In the sentence: “there is a lack of studies that explain the relationship between 80 acculturation, adaptation, SOC, perceived health, and QoL” (line 80-81), it is not clear for me what do you mean by acculturation as a variable, and then “adaptation”, as you can not be more or less acculturated, as you have presented you can present one strategy for acculturation, and Berry’s model defines one as “adaptation”. You need to explain this better in your manuscript.
The conclusion says “barriers accessing health services, improve the delivery of preventive care and provide a basis for health policies to guarantee the appropriate healthcare for a growing number of Croatian migrants” (399-402), but this is completely lacking in the introduction of the manuscript, it seems more a cross-cultural psychology study rather than a public health one. I highly recommend re-framing the introduction with more information about migrants’ barriers to healthcare and health needs in Europe.
Three recommendations coming from an EU project where health status, accessibility, and care delivery for migrants in the EU are discussed:
- Lebano, A., Hamed, S., Bradby, H. et al. Migrants’ and refugees’ health status and healthcare in Europe: a scoping literature review. BMC Public Health 20, 1039 (2020). https://doi.org/10.1186/s12889-020-08749-8
- Riza, E.; Karnaki, P.; Gil-Salmerón, A.; Zota, K.; Ho, M.; Petropoulou, M.; Katsas, K.; Garcés-Ferrer, J.; Linos, A. Determinants of Refugee and Migrant Health Status in 10 European Countries: The Mig-HealthCare Project. Int. J. Environ. Res. Public Health 2020, 17, 6353. https://doi.org/10.3390/ijerph17176353
- Gil-Salmerón, A.; Katsas, K.; Riza, E.; Karnaki, P.; Linos, A. Access to Healthcare for Migrant Patients in Europe: Healthcare Discrimination and Translation Services. Int. J. Environ. Res. Public Health 2021, 18, 7901. https://doi.org/10.3390/ijerph18157901
Reviewer 2 Report
The manuscript entitled "Acculturation, adaptation and health among Croatian migrants in Austria and Ireland: a cross-sectional study" can be considered preliminary study, since a small number of Croatian migrants participated. There are many issues that must be resolved before the manuscript will be reconsidered for publication.
1. The conceptual framework presented on figure 1 is confusing. Introduction does not include any information about sociodemographic "factors" determining SOC and acculturation, therefore the use of these measures in the study is not justified at all. Please add all relevant information about how age, sex, education, living situation (relationship status?), and net income influence both SOC and Acculturation. It is also unclear how acculturation determine health-related behavior. The Authors showed several mediators in the model, but not explained without theoretical justification for these associations. Mediators in the study are: health behavior (for association between SOC and QOL, and between acculturation and QOl), adaptation (for adaptation-QOL association), acculturation (for sense of coherence-QOL association). Moreover, the model was not examined in the study (this is a structural model, which should be tested by path analysis), since regression was used for some associations. Therefore, the results section is inconsistent with the "theoretical model" (which is unjustified by literature presented in the Introduction) and confused.
2. The differences between migrants from Austria and Irleand is not described in the Introduction, so it is unclear, why the Authors expect differences. The specific hypothesis is not formulated, so it is surprising that country differences are examined in the results section.
3. The basic criterion to use parametric tests is normal distribution, but this is not commented in the study at all. It seems unlikely that normality of samples was demonstrated for each variable. In that case, a non-parametric tests should be performed.
4. Nothing is known whether criteria were fit for regression model.
5. Neither power analysis nor effect size (e.g., Cohen's d for t-test) are reported in the study.
6. It is unclear, why the authors claim the CFA showed "good fit". Considering RMSEA, good fit values are less than 0.05. Please add the literature to justify the claim. According to scientific standards, no one variable demonstrate "good fit" in this study. It is also unclear, how many factors were considered in each CFA model. it must be explained.
7. Why Cronbach's alpha is not reported for SLIQ?
8. The Authors performed a series of regressions instead of one path model, which controlled for collinearity and all associations. This procedure is inappropriate and produces a large bias in results. Therefore, the path is required to examine the "conceptual framework" presented on Figure 1. However, if the Authors would like to examine particular dimensions of QOL, SEM with latent variables are necessary. Meanwhile, the current results are inappropriate.
Round 2
Reviewer 1 Report
Thank you very much for addressing all the comments.
Author Response
Thank you very much for your positive review.
Reviewer 2 Report
The authors responded to a selected question and corrected some issues. Unfortunately, the revision showed new evidence against publication of this manuscript.
1. According to the article of Hu and Bentler (1999), to which the Authors refers, the minimal cut-off for RMSEA is 0.06, while for TLI and CFI is 0.90. As such, any one of the variables (SLIQ, SSS-8, BAOS, BPAS, BSAS) has "an adequate fit", as the Authors still claim. In addition, Cronbach's alpha for SLIQ is below an acceptable level. Unfortunately, these flaws can be an effect of many factors, including a bad translation process. As such, all results in the manuscript are unbelievable, as a huge measurement error, and further revisions is unable to change it.
2. There are scientific standards that allow parametric statistics. For example, to use Pearson's correlation, Student-s t-test, and linear regression, normal distribution must be presented. If not, a non-parametric equivalence test can be conducted instead. In addition, as suggested in my previous review, several tests must be performed to check assumptions for the linear regression (e.g., linear relationship, multivariate normality, no or little multicollinearity, no auto-correlation, homoscedasticity). Also, as suggested before, multiple comparisons are related to error and bias, which affect the results significantly. Path analysis would be a better solution in this case.
Hu, L.T.; Bentler, P.M. Cut-off criteria for fit indexes in covariance structure analysis: Conventional criteria versus new alternatives. Struct Equ Model. 1999, 6, 1–55.
